# Thermal Energy Storage by the Encapsulation of Phase Change Materials in Building Elements—A Review

**DOI:** 10.3390/ma14061420

**Published:** 2021-03-15

**Authors:** José Luis Reyez-Araiza, Jorge Pineda-Piñón, José M. López-Romero, José Ramón Gasca-Tirado, Moises Arroyo Contreras, Juan Carlos Jáuregui Correa, Luis Miguel Apátiga-Castro, Eric Mauricio Rivera-Muñoz, Rodrigo Rafael Velazquez-Castillo, José de Jesús Pérez Bueno, Alejandro Manzano-Ramirez

**Affiliations:** 1División de Investigación y Posgrado, Facultad de Ingeniería, C. U. Cerro de las Campanas, Centro, Universidad Autónoma de Querétaro, Querétaro 76010, Qro., Mexico; araiza@uaq.edu.mx (J.L.R.-A.); marroyoc@uaq.mx (M.A.C.); jc.jauregui@uaq.mx (J.C.J.C.); rodrigo.velazquez@uaq.mx (R.R.V.-C.); 2Centro de Investigación en Ciencia Aplicada y Tecnología Avanzada, Instituto Politécnico Nacional, Unidad Querétaro, Cerro Blanco No. 141, Colinas del Cimatario, Querétaro 76090, Qro., Mexico; jpinedap@ipn.mx; 3CINVESTAV Querétaro, Libramiento Norponiente 2000, Fraccionamiento Real de Juriquilla, Querétaro 76230, Qro., Mexico; jm.lopez@cinvestav.mx; 4Departamento de Ingeniería, Universidad de Guanajuato, Guanajuato 36000, Mexico; ragatsi99@yahoo.com; 5Centro de Física Aplicada y Tecnología Avanzada, UNAM, A.P1-1010, Querétaro 76000, Qro., Mexico; apatiga@unam.mx (L.M.A.-C.); emrivera@fata.unam.mx (E.M.R.-M.); 6Centro de Investigación y Desarrollo Tecnológico en Electroquímica (CIDETEQ), Querétaro 76703, Mexico; jperez@cideteq.mx

**Keywords:** phase change materials, construction elements, heat storage

## Abstract

The energy sector is one of the fields of interest for different nations around the world. Due to the current fossil fuel crisis, the scientific community develops new energy-saving experiences to address this concern. Buildings are one of the elements of higher energy consumption, so the generation of knowledge and technological development may offer solutions to this energy demand, which are more than welcome. Phase change materials (PCMs) included in building elements such as wall panels, blocks, panels or coatings, for heating and cooling applications have been shown, when heating, to increase the heat storage capacity by absorbing heat as latent heat. Therefore, the use of latent heat storage systems using phase change materials (PCMs) has been investigated within the last two decades. In the present review, the macro and micro encapsulation methods for construction materials are reviewed, the former being the most viable method of inclusion of PCMs in construction elements. In addition, based on the analysis of the existing papers on the encapsulation process of PCMs, the importance to pay more attention to the bio-based PCMs is shown, since more research is needed to process such PCMs. To determine its thermophysical and mechanical behavior at the micro and macro levels, in order to see the feasibility of substituting petroleum-based PCMs with a more environmentally friendly bio-based one, a section devoted to the excellent PCM with lightweight aggregate (PCM-LWA concrete) is presented due to the lack of description given in other reviews.

## 1. Introduction

One factor that influences the increase in electrical energy consumption globally is the increase in the world population. It is estimated that, in most developed countries, between 20% to 40% of total electrical energy consumption is attributed to commercial and residential buildings, whereas about 50% is associated with the thermal conditioning of interior spaces [1]. The strategies to reduce these high rates of electrical energy consumption have various aspects [2,3]. Most of these attack the problem by increasing the efficiency of thermal conditioning equipment [4]. Other strategies try to develop materials to promote the hermeticity of the spaces to be conditioned [5]. Recently, various fields of engineering have developed research considering “Phase Change Materials” (PCM). These are related to the physical phenomena of phase change and latent heat (energy that is absorbed or released by the phase change in a material) [6,7,8]. The state transition processes (solid–liquid and liquid–solid) related to the fusion and crystallization of a materials are well illustrated by Ventolà, L., M. et al. [7].

A phase change material must have two basic requirements: a suitable phase change temperature and a large melting enthalpy (to achieve high storage density compared to sensible heat storage). However, depending on the application, more physical, technical and economic requirements must be satisfied with adequate functioning in buildings’ conditioning. The main requirements are reproducible phase change (cycling stability) to avoid “phase separation” due to the cycling instability. Little subcooling (temperature significantly below the melting temperature) guarantees that solidification and melting can continue in a minimum temperature range; good thermal conductivity allows latent heat to be stored or released with sufficient heating or cooling power.

Since PCMs repeatedly change phase from solid to liquid, they need to be kept in matrices. For the design of such for PCMs, these must cover the following technical requirements (mechanical, chemical, safety constraints, economics and some others): low vapor pressure, small volume change, chemical stability of the PCM, compatibility of the PCM with other materials, non-toxic, non-flammable materials, large lifetime, abundant and available and cost-effective [9,10,11,12,13,14]. How the PCMs are contained depends on the thermal storage application.

A reliable design for a thermal storage system for any application must also consider the level of purity offered by each PCM manufacturer; otherwise, variations in thermophysical properties may arise in materials that include PCMs [15,16]. Apart from the PCM, the latent heat thermal storage system must contain two or more components to function correctly: (a) a vessel for the PCM (encapsulation of PCMs); (b) a heat exchange surface required to transfer the heat from the heat source to PCM and from PCM to the heat sink. The thermal energy storage systems can be sensitive to either heat storage or latent heat storage, or a combination of both and the storage capacity of the material depends on both its specific heat and latent heat values to obtain an adequate process of the phase change process as solid–solid, solid–liquid, solid–gas and liquid–gas.

Several research works and papers have reported the following ways or process of integration of PCMs in building materials and structures to prevent the escape of PCMs in the melted state: (a) micro-encapsulation (PCMs enclosed in a small capsule); (b) macro-encapsulation (PCMs hermetically closed in a container); (c) shape-stabilized PCMs; (d) impregnation of porous building materials [17].

Hence, from the analysis of results mentioned in previous papers, it may be observed that the main disadvantage of incorporating PCMs by impregnation in porous building materials (stone, wood, concrete, among others) is difficult to hold a PCM in a matrix that is in the liquid state since some organic-PCMs can evaporate or release some of their compounds. In addition, salt hydrates are particularly unsuitable for this way of containment because of their water content. In the particular cases of the mixtures in a base of Portland cement and alkali-activated materials (i.e., geopolymer), the incorporation of PCM may interfere with the cement hydration process causing detrimental effects on the mechanical properties [18,19,20,21].

Concerning the macro-encapsulation process, the PCM is not mixed with the raw material (concrete, gypsum, among others), several papers mention a suitable method to incorporate PCMs in building elements (especially prefabricated walls and roofs). In addition, the PCM microcapsules or shape-stabilized PCMs could be employed to prepare macro-encapsulated PCM members. PCM impregnation in lightweight aggregates is sometimes referred to as macro-encapsulation. However, this process can be attributed to another encapsulation technique [21,22,23,24,25,26,27].

The advantage of this confinement is that the PCM-filled micro-capsules can be mixed with other materials (for instance, concrete, gypsum plaster, geopolymer and wood-plastic composite). The micro-capsule layer can provide a physical barrier between the core material and other components of the products. PCMs need to be capsulated with a hard and more elastic shell when used with building materials [28,29,30,31,32].

However, from the analysis of the information presented in the previously mentioned papers, it may be indicated that the disadvantage is the low thermal conductivity of most materials used for micro-capsules and the small total heat storage capacity due to the relatively small mass-fraction of PCM. The concrete, mortar and plaster must meet specific mechanical properties requirements and include a large number of microcapsules can reduce such properties [33].

Regarding the shape-stabilized PCMs, the stabilizing is achieved using a porous supporting material with high thermal conductivity. In different investigations, the following materials have been suggested: graphite powder, silica fume, bentonite, diatomite and kaolin. In addition, they present three processes to obtain the shape-stabilized PCMs (direct absorption, vacuum impregnation and sol-gel methods). Such manufacturing methods have the characteristic that the molten PCM is absorbed into the host material’s porous structure. A vacuum impregnation method is an excellent alternative to have the most significant amount of molten PCM absorbed in the porous structure and the sol-gel process is a technique for fabricating both organic and inorganic shape-stabilized PCMs [34,35,36,37,38,39,40,41,42,43,44,45,46,47,48,49,50,51,52,53,54].

The graphite powder and silica fume have been often used due to their higher absorption capacity and better applicability to combine into cementitious composites. Silica fume has traditionally been used as a micro-filler pozzolanic addition in cementitious composites (mortar and concrete). Nevertheless, various works have reported that shape-stabilized PCMs may be added to reduce the compressive strength and elastic modulus of cement, mortar, concrete and pavement materials. Several researchers also applied different shape-stabilized PCMs composites in non-structural building elements, such as walls and building envelopes in both laboratory and real scale samples, to regulate indoor temperature variations [55,56,57,58,59].

Today, in any discipline of science and engineering, it is essential to develop environmentally friendly materials. In PCMs, various research works have studied the relevance of using biomass feedstocks which could be transformed into high value-added products for PCM materials. For example, Ryms et al. [59] analyze the possibility of using a new carrier for phase-change material obtained from waste biomass pyrolysis decomposition. In this case, biomass char is added as a PCM carrier. The biomass came from the fruit of inedible chestnut (*Aesculus hippocastanum*), although the char’s general source might originate from any waste biomass [60,61]. This work reports the process of obtaining the char and the preliminary thermal study at the laboratory level. The study consisted of getting the PCM’s thermodynamic properties, composed of char and Rubitherm RT22 pure PCM, and such properties were compared with a Micronal 5040X micro-encapsulated PCM. Looking at the results, it is quite interesting to see that char as a PCM carrier could be beneficial from a thermodynamic perspective and could serve as an alternative to commercial products. However, tests have not yet been carried out integrating this bio-based PCM with construction materials to corroborate the mechanical and thermal properties.

Other works investigate the thermo-physical properties of different organic bio-based PCMs, Nazari et al. [62] contrasted the thermal behavior of three paraffin-based PCMs (RT24, RT25 and RT26 by Rubitherm GmbH, Germany) and one bio-based PCM (PureTemp25 by PureTemp LLC, Minneapolis, MN, USA). Fabiani et al. [63] used fat wastes from slaughterhouse residues without further chemical processing to produce PCM composites. This work used thermogravimetric analysis techniques (TGA) and differential scanning calorimetry (DSC) to evaluate their thermophysical performance and determine their thermal energy storage capacities. According to the results presented in these papers, the feasibility of substituting petroleum-based PCMs with a more environmentally friendly bio-based one is shown [64]. However, it is worth indicating that before they can be available for practical applications, the mechanical properties of construction materials that include bio-based PCMs should be evaluated in terms of thermal properties. Moreover, these research works can contribute knowledge to establish standards to characterize and analyze organic PCMs.

Studies on the use of PCMs for thermal energy storage in building applications have been carried out since 1970. Furthermore, as a difference to our review, several review papers published since then are focused on the analysis the analysis and discussion of the thermal performance of various types of systems such as PCM wallboards (gypsum, among others), PCM shutters, PCM building blocks, air-based heating systems, floor heating and ceiling boards. In general, they report having used commercial PCM materials and describe recent efforts to develop new phase change materials. Different discussions have been focused on methods of thermal properties characterization, long term stability and encapsulation and heat transfer by numerical simulation techniques and experimental techniques [65,66,67,68,69,70,71,72,73,74,75,76,77,78,79,80,81].

Concerning some recent experimental studies in real climatic conditions or using environmental chambers to evaluate the performance of PCM-incorporated in building elements, the following building elements have been reported:small house and room model;test huts;concrete wallboard sample;masonry brick wall;concrete sandwich panel walls;concrete core slab in test cubicles;pavement.

In these building elements, the PCMs were included in various ways (for instance, macro-encapsulation, PCM in layer and multi-layer walls, PCM underfloor heating system, hollow steel balls macro-capsules, PCM micro-capsules and porous inclusion). The targets were different, including thermal performance evaluations, validation of numerical simulations, energy performance and cost savings, thermal stress control in concrete and ice/snow-melting [77,78,79,80,81,82].

However, taking into account the previously mentioned papers, it may be inferred that there are few studies on real-scale models and even these only focus on studies on thermal behavior (without hesitation, the most relevant aspect of PCM materials; otherwise, it would not make sense to include them in construction elements). However, if the systems with PCMs meet the thermal requirements, studies must be carried out on the structural behavior of buildings with PCM under quasi-static and dynamic monotonic loads to ensure that they can safely be used to propose standards and building regulations with such systems.

In contrast, other studies on cementitious materials have pointed out that the inclusion of PCMs in laboratory specimens can mitigate the temperature cracking of concrete. Research work has analyzed the variation of temperature and the heat flow in concrete elements during Portland cement’s hydration process. Likewise, the damage due to freeze–thaw cycles in concrete pavements incorporating PCM has been studied [83,84,85]. The elements of envelopes (clay partitions, panels, or lightweight composite material walls) are among the most recurrent in research, developed by combining cementitious materials with PCMs. Research has demonstrated the benefit of including PCM’s materials, evaluating thermal behavior directly on full-scale models with and without PCM materials. In some cases, a minimum reduction of approximately 2 °C has been obtained in the maximum temperature reached [86,87,88,89,90].

In summary, so far in the present article, the large numbers of PCMs that melt and solidify at a wide range of temperatures are mentioned. The aggregate methods for construction materials are reviewed, emphasizing micro and macro-encapsulation, being the latter the most viable way to include PCMs in construction elements. The methodology and processes developed to achieve this confinement are described. Moreover, based on the papers revised critically, this article emphasizes the need to investigate the processes of the bio-based PCMs and the thermophysical and mechanical behavior at the micro and macro level to demonstrate the feasibility of substituting petroleum-based PCMs with a more environmentally friendly bio-based one. It is also vital to obtain multi-physics models of the nonlinear behavior of materials with PCMs to understand their behavior better and design optimized systems. Finally, a section for PCM with lightweight aggregate (PCM-LWA concrete) is presented. In contrast to and to strengthen the differences with other reviews available in the literature, it is worth mentioning that in the literature, several reviews available are dedicated to thermal transfer and energy storage [5,6,15,71,78] and others reviews like: Cabeza L.F. et al. covers shortly either micro or macro-encapsulation along with long term stability and thermophysical properties [74], the reviews from Jamekhorshid A. et al. [91] and Konuklu Y. et al. [75] cover just microencapsulation methods, Marani A. et al. [76] examines potential methods of incorporating PCMs into building materials (micro and macro-encapsulation), with an emphasis in applications and stabilization materials and methods in concrete, while the Rathore P. K. S. and S. K. Shukla review [77] comprises the macro-encapsulation technique along thermal energy storage methods used in buildings. Meanwhile, Tung-Chai Ling and Chi-Sun Poon’s review [92] describes the incorporation techniques, immersion and impregnation techniques in concrete.

## 2. General Classification of PCM Materials, Comments

Materials for phase change thermal energy storage must have a large latent heat and high thermal conductivity. They should have a melting temperature lying in the practical range of operation, melt congruently with minimum subcooling, be chemically stable, low in cost, non-toxic and non-corrosive. The PCMs are grouped into organic, inorganic and eutectic mixtures of compounds that can yield different phase change temperatures. This classification has emerged from studies carried out over 40 to 50 years and, in that period, other authors have shown the advantages and disadvantages of PCMs. Different experimental techniques have been reported and used to determine these materials’ behavior in melting and solidification. Hence, according to the analyzed papers [6,7,8,9,10,11,12,13,68,69,74,75,78,90,91,92,93,94,95,96,97], it is concluded that no material has all the optimal characteristics required for a PCM and selecting a PCM for a given application requires careful consideration of the properties of various substances.

Differential scanning calorimetry methods (DSC) are widely used to determine the thermophysical properties (heat of fusion, specific heat and melting point) of PCMs and have served to classify them. However, it may be suggested that other methods [95,96,97,98,99,100,101,102,103,104,105,106,107,108,109], such as the T-history method and conventional calorimetry methods, have also been used. The analysis of these properties is carried out with thermograms, in which we can identify the values of the phase transition temperatures during melting and freezing. However, as tiny samples are used to determine these properties’ values, they may vary with large samples.

### 2.1. Organic

Three sub-types can be highlighted within the organic phase change materials: paraffins, fatty acids and polyethylene glycol (PEG). In general, these compounds have good thermal properties and convenient chemical stability [16]. As a result, it is seen how much of the research is currently related to different kinds of paraffin’ uses in construction elements for buildings because their chemical stability facilitates the synthesis of materials and products’ manufacture.

#### Organic No-Paraffinic Materials

Two major categories of organic PCMs are paraffin and non-paraffinic materials. Because of their high flammability, most paraffins have not been used in PCMs. However, using a suitable container paraffin has been widely used due to their large latent heat and good thermal characteristics (minimal supercooling, varied phase change temperature, low vapor pressure in the melt, good thermal and chemical stability and self-nucleating behavior) [62]. In contrast, the literature indicates how the bio-based (non-paraffin) PCMs are significantly less-flammable than paraffin. Bio-based PCMs are organic fatty acid ester PCMs made from underused and renewable feedstock, such as vegetable oils, animal fats, industry or agricultural wastes, mainly composed of highly sustainable bio-sources.

Several researchers [61,107,108] have focused on these bio-based (non-paraffin) PCMs, for testing their suitability in thermophysical properties and heat storage capacity in terms of phase-change enthalpy, specific heat capacity and melting temperature. However, only limited thermal data are now available in the literature. Furthermore, there are no norms or standards to characterize Bio-based PCMs.

### 2.2. Inorganic

Compared to organic PCMs, there are fewer inorganic compounds that can be used in the construction industry. The two most widely used types are hydrated salts and metallic salts, where hydrated salts have been the most studied in all fields of research related to PCM’s [73].

### 2.3. Eutectics

Eutectic PCM is a mixture of two or more PCMs to achieve a desirable melting point. Eutectic nearly always melts and freezes without segregation because the components are selected to freeze and melt simultaneously. These PCMs acquired great importance due to various eutectic compounds with different properties that can benefit thermal energy storage systems. In general, from the literature [106,107,108,109,110], it is clear that non-organic PCMs have better thermal storage characteristics; however, they tend to be more expensive than paraffin.

Table 1 shows some of the chemical components used for construction materials. Although these properties are the main ones, it is convenient to consider other secondary properties, which can be very important depending on the application, geographical area and number of cycles. The thermal properties of PCM suitable for buildings is described by Khudhair, A. M. and M. M. Farid [6], Memon, S. A. [109] and Cui, Y., J. Xie, J. Liu and S. Pan [110]; meanwhile the advantages and drawbacks from various PCMs are indicated in Khudhair, A. M. and M. M. Farid [6], A. Felix Regin et al. [15], Baetens, R., B. P. Jelle and A. Gustavsen [107] and Cui, Y., J. Xie, J. Liu and S. Pan [110].

## 3. Encapsulation Processes for PCMs

A successful PCM thermal storage system should have a suitable container for the PCM (encapsulation of PCMs) and a heat exchange surface for transferring heat from the heat source to PCM and from PCM to the heat sink. It is necessary for proper operation that the encapsulation has strength, flexibility, corrosion resistance, thermal stability, structural stability and easy handling. Bulk storage in a tank (tank heat exchangers with more extensive heat transfer), macro-encapsulation and microencapsulation are the types of confinements most analyzed for PCMs.

### 3.1. Macro-Encapsulation

The most common type of PCM containment is macro-encapsulation, in which a significant quantity of PCM is encapsulated in a discrete unit (encapsulation in containers usually larger than 1 cm in diameter). The shape of the encapsulating shell can be any form (tubes, cylinders, pouches and cubes). The most cost-effective containers are plastic bottles (high density and low-density polyethylene bottles and polypropylene bottles), tin-plated metal cans and mild steel cans. The mass of PCM per unit may range from a few grams to a kilogram. The analysis of the revised papers [107,108,109,110,111,112,113,114,115,116,117,118] shows how macro-encapsulated PCM can be easily prepared in any shape and size to suit different applications. Unlike micro-encapsulation, where various methods and techniques are used to encapsulate the PCM, the PCM’s macro-encapsulation does not require any pre-defined process.

By careful selection of the capsule geometry and the capsule material, the macro-encapsulation can be used for a wide variety of energy storage needs and can be to get incorporated easily into the building envelope of any shape, size and dimension. However, the compatibility of the shell material with building material and PCM is the area that still needs investigation for further improvement [107,108,109,110,111,112,113,114,115,116,117,118]. The advantage of the macro-encapsulation is its applicability to both liquid and air as heat transfer fluids and easier to ship and handle. In general, macro-capsules are incorporated on exterior walls and precast slabs since buildings’ facades are in direct contact with weather conditions as well as to expose to solar radiation. Its operation depends on several factors:the location of the macro-capsule (the internal or external surface or within the construction element);the local weather conditions of the region (ambient temperature, solar radiation);the geometric characteristics of the construction part (concrete wall, partition masonry);the conductive characteristics of heat through the building material and the type of PCM.

Some reputable macro-encapsulated PCM manufacturers have been prepared and marketed as various forms of microencapsulates, such as Microtek Laboratories [119], Rubitherm technologies [120], Pure temp LLC [121], Shanghai Tempered Entropy New Energy Co. [122], MikroCaps [123], Winco technologies [124] and Teappcm [125], which individually market the following products: polymers, aluminum panels, polymer bags, pouches, aluminum tube, plastic blocks, breather membrane, stainless steel balls and HDPE (High Density Polyethylene) panels. Manufacturers of products ready for building applications are shown in Baetens, R., B. P. Jelle and A. Gustavsen [107], Kalnaes, S. E. and B. P. Jelle [126] and Vicente, R. and T. Silva [127], while the characteristics of macro-encapsulated PCMs for buildings are summarized in Table 1.

### 3.2. Microencapsulation

Microencapsulation refers to techniques in which small PCM particles or droplets (spherical or rod-shaped) are contained within a sealed, continuous shell (thin and high molecular weight polymeric film or inorganic film). This technique is an excellent solution to prevent leakage of the melted PCM in latent heat thermal energy storage systems, reduce PCM reactivity with the outside environment. The critical analysis of the literature [126,127,128] shows how micro-encapsulation improves heat transfer to the surrounding through its large surface to volume ratio and improves cycling stability since phase separation is restricted to microscopic distances; however, the cost of the microencapsulation system is high compared to other thermal storage methods. The coated particles can then be incorporated in a powder form or dispersed into a carrier fluid in any matrix compatible with the encapsulating film. It follows that the film must be compatible with both the PCM and the matrix [126,127,128].

The possible morphologies (shapes) of microcapsules can be diverse (irregular shape, spherical, tubular, multi-wall, multi-core and matrix particle) and depending on the arrangement of the core material and the deposition process of the shell. There can be four types of distribution: mononuclear (core/shell), polynuclear (many cores coated with a continuous shell material), matrix encapsulation (the core material is distributed homogeneously) and multi-film (a continuous core coated with multilayer continuous shell material) [129,130,131].

Organic PCM core has a suitable melting point near the thermal comfort range of humans around 20 °C and has several advantages over other types of PCM materials. Organic PCM includes different classes like paraffin (n-alkane), fatty acids, alcohols, esters and polyethylene glycol (PEG). Paraffin class materials of organic PCM are the most popular choice as core materials. On the other hand, polyethylene glycol (PEG) is difficult to encapsulate. Similarly, inorganic salts are also rarely encapsulated due to their solubility in water.

Shell materials form the capsules that contain the PCM and can be made from an organic or inorganic material or hybrid shell materials made of organic-inorganic combination. The majority of the shells are organic and prepared through chemical methods like polymerization. A shell material should not undergo a chemical reaction with the PCM core and should possess good chemical and thermal stability. Its surface morphology must be smooth hand it should have minimum porosity and prevent any leakage of PCM at temperatures above the melting point of PCM. The shell provides mechanical strength (so that thicker layers show a much better mechanical behavior) and also shape stability, and it is desirable to have shell material with high thermal conductivity.

Concerning the encapsulation process, from the literature, it may be pointed out that there are three different methodologies to microencapsulated PCM, and the most appropriate technique depends on the physical and chemical properties of the materials to be used [33,130,132,133,134]:
(a)physical methods: pan coating, air-suspension coating, centrifugal extrusion, vibrational nozzle, spray drying and solvent evaporation;(b)physic-chemical methods: Ionic gelation, coacervation, sol-gel;(c)chemical methods: interfacial polymerization, suspension polymerization, emulsion polymerization.

Research reports [16] mention that the mean diameter, the thickness of the shell and the mass percentage of PCM compared to the total mass of the capsule is related to the quality of the microencapsulation process PCMs. This methodology for PCM still needs to be improved because the microcapsules can break when colliding with other microcapsules when used in active systems. Elsewhere, integrated carbon additives in building materials made of composite materials with PCM microcapsules have reported improvements in efficiency and the heat transfer rate [135,136]. In summary, in the literature revised so far, we can observe that many researchers have studied microencapsulation, but the literature is scattered [136,137].

With micro-encapsulated phase change materials products for buildings, two companies (BASF and Microteklab) have designed micro-capsules for home and office applications; furthermore, Rubitherm GmbH (Germany), Cristopia (France), TEAPEnergy (Australia), PCMProducts (UK), Climator (Sweden) and Mitsubishi Chemical (Japan) already offer commercial PCMs.

One of the objectives of using PCMs in mortars and concretes, regardless of the method of incorporation, is to improve their sustainability and durability [137,138,139,140]. Wei et al. [141] evaluated Portland cement-based mortars’ durability with micro-capsules manufactured by Microteklab and Micronal-BASF. Their results showed a reduction of approximately 25% in the phase change enthalpy. Such a decrease was not ascribed to the microcapsules’ mechanical affectation during the mixing process, but the chemical reactions with the sulfate ions. Moreover, this study shows how the chemical reaction between PCM and ions does not affect the durability of the mortar.

## 4. PCM with Lightweight Aggregate

Another interesting method is observed in the literature available so far for including PCMs in the building materials is incorporating PCM with lightweight aggregate (PCM-LWA). PCM can be added to concrete by shape-stabilized PCMs, direct incorporation and immersion and encapsulation. This PCM compound’s manufacturing technique introduces the PCM into the lightweight aggregate (pumice, perlite, expanded shale/clay, expanded slate, expanded perlite, expanded vermiculite.). This technique is comparable to the shape stabilizing method, with some notable distinctions. In this case, the support material is the lightweight aggregate; in contrast, the shape-stabilized PCMs are used powder materials (fine aggregates), such as graphite powder or silica fume [25,26,142,143,144,145].

Vacuum impregnation and direct impregnation are the methods used to obtain PCM-LWA composite. In vacuum impregnation, a pump is used to remove the air from the pore of LWA, enabling better absorption capacity of aggregates. Memon et al. [26] reported that the absorption capacity of LWA using vacuum impregnation was 74%, in contrast with 18% using direct immersion. However, analyzing this process seems complicated and time-consuming; this technique is generally considered impractical [146].

There are some essential factors to obtain this PCM-LWA and for it to be considered as a constituent in mixing proportions for mortars or concretes, such as the type of lightweight aggregate and its absorption capacity (porosity, pore size, aggregate size and surface area, temperature and viscosity of liquid PCM), the method to impregnate, the materials for coating and support, as well as characterization and performance testing.

Numerous researchers [25,26,27,45,47,81,142,144,146,147,148,149,150,151,152,153,154,155] have proposed several porous aggregates as PCM hosts, distinct impregnation processes and various coating and support materials. It was observed that porosity is not the only measurable parameter for the PCM absorption capacity of LWAs. In addition, the pore and aggregate size can affect this property. Nonetheless, Aguayo et al. [146] took the surface adsorption of small particles into account and suggested removing particles finer than 150 mm from LWAs to prevent surface adsorption of PCMs in place of its absorption into the LWAs pores. Moreover, researchers have proposed adding supporting materials and other coatings (cement paste, silicone coating, bituminous emulsion, epoxy resin, graphite powder, silica fume) to moderate leakage molten PCM and make thermal and mechanical properties of the PCM better. In mixture proportioning with Portland cement, the gradual scape of PCM from pores of LWAs can make a difference to cement hydration reactions and consequently affect the compressive strength of concrete [25,26,27,45,47,81,142,144,146,147,148,149,150,151,152,153,154,155].

Regarding the covering placed on the PCM-impregnated porous material particles, it has been found that a low thermal conductivity of this can induce a short performance of the latent heat of the phase change within the encapsulation. Therefore, two study approaches emerge to take advantage of the thermal storage capacity: (a) determine the optimal thickness and area with which the aggregate should be covered, both to reduce damage in the mixing process and improve thermal transfer; (b) use smaller particles on the cover. Calvet et al. [156] and Wang et al. [157] observed a reduction in loading and unloading thermal energy time without affecting the capacity to store energy when graphite was incorporated in the PCM. However, these results are not wholly replicable for mortar mixtures with particles of 5 to 10 mm in diameter.

Other works such as that of Dumas et al. [158], Joulin et al. [159] and Sierra [160] report studies on the thermodynamic and thermal transfer processes of materials with PCMs. Likewise, research such as that of Haurie et al. [161,162] reports the flammability disadvantage of paraffin-based PCMs used in mortars and concretes under fire, despite their thermal advantages over hydrated salt-based PCMs. The work of Sierra [160] shows a differential calorimetry analysis of an organic PCM composed of butyl stearate and soy wax, which was vacuum impregnated in crushed pumice particles. In addition, graphite powder was added to promote thermal conductivity. The work’s main objective was to evaluate the effect of thermal conductivity and latent heat stored in samples with different percentages of graphite powder when they are subjected to 100 work cycles with temperature variations using Peltier cells. The analysis showed that organic PCM-LWA stores heat energy.

Several investigations [25,26,27,142,144,146,147,148,149,150,151,152,153,163,164,165,166] have already carried out microstructural analysis (SEM, DSC, TGA and FTIR) of PCM-LWA and analysis of the thermo-mechanical properties (thermal performance, thermal conductivity, compressive strength, shrinkage strain and effects of freezing and thawing cycles) of concretes and mortars incorporating PCM. No matter the method of mixing PCMs in concrete, most of the research argues that it lowers resistance to compression. Elsewhere [38,167,168,169,170,171,172], it has been reported that when the encapsulation techniques for plain concrete are compared with others, the reduction in compressive strength of concrete incorporating microcapsule may be ascribed to two factors: the significant disparity between the intrinsic strength of the microcapsules and the concrete constituents and damage of microcapsules during mixing resulting in leakage of PCM. Due to these reasons, macro-encapsulation with a strong shell is preferred.

As a result, it may be inferred that more research is needed to obtain a structural PCM-LWA concrete with useful mechanical, thermal and heat storage properties, along with an understanding of their effects on the thermal and mechanical behavior of PCM-LWA with different coatings and supporting materials.

## 5. Discussion and Comments

Based on the review we made of the publications cited in this work [15,16,20,67,68,69,70,71,72,73,74,75,76,77,78,87,94,105,107,108,124,126,127,128,130,139,140], we can mention that there is still no in-depth discussion for creating a multi-physical model that couples mechanical behavior with thermal behavior for construction systems with PCM. Both behaviors depend on the boundary condition related to the contact phenomenon, which models bodies’ interaction across their boundaries. Contact is a non-linear phenomenon that relies on adherence (chemical, friction, mechanical anchoring), surface roughness, contact surface and are considered local phenomena that affect the bodies’ total behavior in contact (level macro). Likewise, both PCM encapsulates (micro and macro-encapsulation) and PCM construction systems in a building interact across their boundaries through the phenomenon of contact. In particular, the articles mention [9,10,11,12,13,14] such an effect as the compatibility between materials and assume the understanding of the concept without discussing it clearly.

Concerning any PCM encapsulation according to its components (cover, PCM base, support material, container), the phenomena of heat transfer by conduction, convection and radiation, determine the capacity to store and release thermal energy in PCMs. All these transfer mechanisms are phenomena that evolve and depend on the bodies’ contact surface. Likewise, the proportionality coefficients corresponding to each heat transfer mechanism (conduction-k, convection-h and radiation-s) also influence the transfer of energy by heat flow; they are experimentally obtained constants and their value depends on the type of material and the roughness of the surface, as well as the radiant energy emissivity of each material.

Contact phenomenon determines the thermal and mechanical behavior of materials with PCM; in other words, the greater the contact area, the more effective contact area and the more uniform transfer of stresses, deformations and loads through the boundaries of the elements of the materials with PCMs. Moreover, since the contact phenomenon influences both the thermal behavior and the mechanical behavior (state of stresses and deformations) of constructive systems with PCM; hence, it is important to obtain and adequately analyze a multiphysics model that couples both behaviors. However, the effectiveness of such a transfer also depends on the continuity of the contact area; in areas close to the discontinuities (holes, abrupt changes in geometry, change of material), stress concentrators are generated, so stress gradients change from one place to another and induce maximum stresses that cause non-linear failures. Unfortunately, in both microencapsulated and macro-encapsulated, there are such discontinuities to affect the mechanical behavior.

The contact phenomenon for micro-encapsulation and macro-encapsulation can have different characteristics. For instance, in macro-encapsulation, there is a more even contact surface (apparently less rough surface) due to the container conditions that retain the base PCM. This fact could ensure a greater contact area between the base PCM and the container’s internal surface; consequently, it is more likely to obtain greater efficiency in transferring energy from the PCM base to the container and the other way round. Concerning the contact between the external surface of the macro-encapsulate and the construction element of the buildings, a greater contact area could not be ensured; as a result, the heat transfer could be less effective. As already mentioned previously, in addition to the contact area, the energy transfer will be more efficient if the conduction, convection and radiation coefficients, corresponding to the container of the PCM base material, and the building element material are optimal for the application. Finally, the internal and external roughness of the materials of PCM systems could affect the heat transfer mechanisms because a rough surface provides a smaller contact area and affects the phenomenon of convection and thermal emissivity and, therefore, the transfer efficiency decreases.

On the contrary, in micro-encapsulation, the effective contact surface can be a function of different variables, such as the geometry of the microcapsule and the amount of these in the construction material (concrete, paste cement, mortar and plaster). Theoretically, the surface area of the microcapsules is more significant if there is a large amount of them in the material, and therefore, the energy transfer must be more effective; however, the effectiveness of the phenomenon can be reduced if the microcapsules are separated by voids or another material with different thermal properties. In the same way as macro-encapsulates, surface roughness and adhesion may affect energy transfer phenomena; however, such physical phenomenon behavior can be more random.

Since microencapsulates are usually mixed with other materials in a fresh state and later set or harden. Chemical adherence is highly desirable in the cover of such encapsulates to ensure compatibility in the system. In addition, the microcapsules can develop mechanical anchoring due to the irregular shape of their surface. Finally, some of the references cited in this article [18,19,20,21,25,26,27,82,83,134,135,136,137,138,139,140,141,163,164,165,166] describe that the addition of microcapsules interrupts the setting processes (hardening or hydration) in Portland cement-based materials, affecting the resistance of materials with PCM. In this sense, it is worth developing microencapsulates that can be compatible with the hydration process in these materials.

According to this, a strategy to make the transfer processes more efficient in PCMs relays in modifying the contact area, roughness and adherence in the construction system with PCM materials. According to the buildings’ application, this is associated with the appropriate choice of the PCM base, cover, container and encapsulation process.

The critical analysis of the literature [126,127,128] shows how micro-encapsulation improves heat transfer to the surrounding through its large surface to volume ratio and improves cycling stability since phase separation is restricted to microscopic distances; however, possibly the cost of the microencapsulation system is high compared to other thermal storage methods. On the contrary, the analysis of the revised papers [109,110,111,112,113,114,115,116] shows that macro-encapsulated PCM can be easily prepared in any shape and size to suit different applications. Unlike micro-encapsulation, where various methods and techniques are used to encapsulate the PCM, the PCM’s macro-encapsulation does not require any pre-defined process.

Now taking about the thermophysical properties (such as heat of fusion, specific heat and melting point) of PCMs, which can be determined by differential scanning calorimetry methods (DSC), are widely used and served to classify the PCMs. However, it may be suggested other methods [95,96,97,98,99,100,101,102,103,104,105], such as the T-history method and conventional calorimetry methods, have also been used. The analysis of such properties is carried out with thermograms, in which we can identify the values of the phase transition temperatures during melting and freezing. However, as tiny samples are used to determine such properties’ values, they may vary with large samples.

In contrast to other reviews, in the current work, the complexity of PCM systems is evident. This article highlights some of the particularities at the micro and macro levels that influence PCM materials’ behavior. Although many research works mention the concept of compatibility and assume a complete understanding of the idea, so far, in the present review, it is detected that there is no consensus to establish a model that represents all the multiphysics phenomena that are coupled in each component of the system (materials) and the entire system of materials for construction with PCMs.

Moreover, there are no standards for construction processes or mechanical and thermal macro and microstructural characterization that give us index parameters of the proper functioning of PCMs. They are new materials, so we rely on other traditional material standards to obtain some mechanical and thermal properties to apply them in the field of engineering.

All new material is obliged to obtain its constitutive equations or behavior rules (each material behaves differently) under different load states to predict service conditions and failure theories to use them in practical situations. Composites with PCM materials are no exception. This is in addition to the complexity outlined above.

In the present review, we want to draw attention to the various research works [60,61,62,63,64,65,107] that have studied the relevance of using biomass feedstocks which could be transformed into high value-added products PCM materials. However, tests have not yet been performed to integrate this bio-based PCM with construction materials to corroborate the mechanical and thermal properties. Likewise, other works investigate the thermo-physical properties of different organic bio-based PCMs, the results of their thermophysical performance led to demonstrate the feasibility of substituting petroleum-based PCMs with a more environmentally friendly bio-based one. However, before using them in practical applications, the mechanical properties of construction materials that include bio-based PCMs should be evaluated.

## 6. Conclusions

Based on the extensive list of references discussed and critically reviewed in the present work about macro and microencapsulation methods for construction materials, the former being the most viable method of inclusion of PCMs in construction elements, it is suggested to pay more attention to the bio-based PCMs. Nevertheless, more research is needed to process such PCMs and determine their thermophysical and mechanical behavior at the micro and macro level to see their feasibility for substituting petroleum-based PCMs with a more environmentally friendly bio-based one. The excellent incorporation of PCM with lightweight aggregate (PCM-LWA concrete) brings another useful alternative to introducing PCMs in concrete. However, again, more research is needed to obtain and understand a structural PCM-LWA concrete with useful mechanical, thermal and heat storage properties knowing their effects on PCM-LWA with different coatings and supporting materials. Hence, the present review may represent a precious indicator of establishing research and development opportunities in techniques and methods for obtaining PCMs and contributing to optimizing methods and new encapsulation techniques to achieve the final product’s desirable thermal and mechanical properties for the cost-effective application of the MPCM with the highest thermal energy storage capacity.

## Figures and Tables

**Table 1 materials-14-01420-t001:** Characteristics of macro-encapsulated PCMs for buildings.

1. Melting temperature	Liquid–solid phase transition temperature close to the required operating temperature range
2. Phase change enthalpy	A high value improves the energy storage density in the system; value close to 200 kJ/kg
3. Specific heat capacity	In general, it should be more than 2.5 kJ/kg °K
4. Thermal Conductivity	High thermal conductivity will improve thermal charge and discharge speed; value greater than 0.6 W/m °C
5. Thermal Cycles	This must be able to experience over 5000 thermal cycles of charge and discharge
6. Over-cooling	This should not undergo over-cooling, because the PCM will not completely solidify below freezing. This could reduce heat removal during freezing
7. Change in Volume	This should experience minimal change in volume during phase change, a large change will increase the size of the container
8. Congruent fusion	Must be completely melted and frozen to ensure homogeneity in the solid and liquid phase. If this is not congruent, it will generate segregation due to the difference in densities
9. Vapor Pressure	You must have a low vapor pressure in the operating temperature range to avoid containment problems
10. Non-corrosive	It must not be corrosive or toxic to the environment
11. Economical and Availability	Must be available on a large scale and at an economical price
12. Non-flammable	Must not be flammable to avoid any fire hazard.

## Data Availability

Data available in a publicly accessible repository.

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
