# Peer review of "Thermal Energy Storage by the Encapsulation of Phase Change Materials in Building Elements—A Review"

_materials, 2021, doi:10.3390/ma14061420_

Round 1
Reviewer 1 Report
This review paper lacks critical analysis of previous works. It just mentioned what has been done in the previous studies without critical analysis and comparison of those works. There was no identification research gap, research need and no mention of future research direction.
Author Response
"Please see the attachment."

Reviewer 2 Report
This paper presents a review of large number of phase change materials (PCM) as a thermal energy storage materials, analyzes methods of micro and macro-encapsulation that can be use in construction elements, and shows the methodology and processes accompanying these topics. Authors provide interesting investigation and present review section particularly about PCM with lightweight aggregate (PCM-LWA concrete) as a PCM carrier.
Although presented article may be interesting to the readership of this journal, the paper may only be considered for publication after the following concerns have been addressed successfully in a revision:
In general:
1) In my opinion the title should be change into more traditional form: "Thermal energy storage by the encapsulation of phase change materials in building elements - A Review".
2) What is a novelty in this article compared to others of this type? Unfortunately I haven't found any new information in here. Authors in my opinion should consider adding a reference to more current state-of-the-art papers (from year 2019 and 2020 for example) related to the subject of the article (please compare with point 3).
Regarding to the "Introduction" part:
3) The introduction chapter in my opinion should have some more references to the current state-of-the-art papers, which are directly related with presented researches, especially those, which analyze the use of PCM and its application in building materials. Although the authors cited interesting considerations regarding the use of the lightweight aggregate saturated with PCM as a temperature stabilizing material, it is worth adding newer reports on this topic, such as: "Possibilities and benefits of a new method of modifying conventional building materials with phase-change materials (PCMs) (from 2019)", or "Post-Pyrolytic Carbon as a Phase Change Materials (PCMs) Carrier for Application in Building Materials. (from 2020)" as a reference. In my opinion extending current analysis with these papers may be interesting for the readers and give a wider scope of view on the pros and cons of applicating PCM.
Regarding to the "Regarding to the "Introduction" part:
4) I recommend Authors to extend this chapter by "other types of PCM carriers" and to include here, for example, new research under biochar as a PCM carrier. This topic could be the mentioned above "novelty" for this manuscript.
After the minor changes described above, the Manuscript may be considered for publication.
Author Response
"Please see the attachment."

Reviewer 3 Report
Please see the attached file.

Author Response
"Please see the attachment."

Round 2
Reviewer 1 Report
The key issue with this paper is the scope. PCM is a big filed with a number of applications, methods and properties. It is not clear what aspect of PCM has been critically reviewed in this paper. The title says encapsulation. So author should provide more emphasis on encapsulation process of PCM, critically review the existing papers on encapsulation process of PCM, what works and what does not work, identify future research need.
The added discussion is very shallow and is not meaningfull.
Author Response
"Please see the attachment."

Reviewer 3 Report
The new section 5 Discussion and recommendations defines partially the rationale behind the work. However, it is strongly recommended to check the English. For example, the term “mechanical witnessing” should be substituted with a standard term.
The objectives of the review article are still not clearly stated, which does not allow the reader to understand what is the novelty this article brings compared to other similar reviews (there are many review articles discussing PCMs, their applications and properties)
Author Response
"Please see the attachment."
